# Chromatin-Mediated Regulation of Genome Plasticity in Human Fungal Pathogens

**DOI:** 10.3390/genes10110855

**Published:** 2019-10-28

**Authors:** Alessia Buscaino

**Affiliations:** Kent Fungal Group, School of Biosciences, University of Kent, Canterbury Kent CT2 7NJ, UK; A.Buscaino@kent.ac.uk

**Keywords:** epigenetics, genome stability, heterochromatin, human fungal pathogens, chromatin modifications

## Abstract

Human fungal pathogens, such as *Candida albicans*, *Aspergillus fumigatus* and *Cryptococcus neoformans*, are a public health problem, causing millions of infections and killing almost half a million people annually. The ability of these pathogens to colonise almost every organ in the human body and cause life-threating infections relies on their capacity to adapt and thrive in diverse hostile host-niche environments. Stress-induced genome instability is a key adaptive strategy used by human fungal pathogens as it increases genetic diversity, thereby allowing selection of genotype(s) better adapted to a new environment. Heterochromatin represses gene expression and deleterious recombination and could play a key role in modulating genome stability in response to environmental changes. However, very little is known about heterochromatin structure and function in human fungal pathogens. In this review, I use our knowledge of heterochromatin structure and function in fungal model systems as a road map to review the role of heterochromatin in regulating genome plasticity in the most common human fungal pathogens: *Candida albicans*, *Aspergillus fumigatus* and *Cryptococcus neoformans.*

## 1. Introduction

Human fungal pathogens are a leading cause of human mortality worldwide and kill almost 2 million people every year [1]. There are only four major classes of antifungal drugs available to treat life-threating fungal infections and the staggering recent escalation in the number of antifungal resistant strains poses an ever-increasing threat to human health and highlights the urgent need to better understand the biology of these microorganisms [2]. *Candida albicans*, *Aspergillus fumigatus* and *Cryptococcus neoformans* are the most common human fungal pathogens responsible for life-threating infections, especially in immune-compromised patients [1].

During colonisation and infection of the host, human fungal pathogens encounter many different host environments to which they must adapt rapidly. For example, they must adapt to grow at high temperature (i.e., fever) and they must survive the dramatic environmental changes following engulfment by macrophages and neutrophils of the host immune system. Indeed, following phagocytosis, fungal pathogens must resist reduced pH and the generation of toxic compounds such as hydrogen peroxide in the microcidal phagolysosomes [3].

Similarly, during development of drug resistance, human fungal pathogens must survive the fungistatic and fungicidal effect of anti-fungal drugs. How do fungal pathogens respond rapidly to the continuous changes in the environment they encounter in the host and following drug treatment? Addressing this question is key to understanding how human fungal pathogens establish life-threating infections and develop resistance to anti-fungal drugs.

In recent years, the importance of stress-induced genome instability has emerged as a key adaptive mechanism of human fungal pathogens [4]. Although excessive genome instability is harmful, moderate genome instability facilitates rapid adaptation to environmental insults. Indeed, genomic instability can increase genetic diversity, thereby allowing selection of genotype(s) better adapted to a new environment (Figure 1). Accordingly, natural isolates of many human fungal pathogens exhibit a broad spectrum of genetic and genomic variations including chromosome rearrangements, short and long range loss of heterozygosity (LOH) events, single nucleotide polymorphisms (SNPs) and whole chromosome aneuploidy [5,6,7,8]. Importantly, genomic instability dramatically increases upon exposure to host-relevant stresses [9,10] and the resulting genetic variation can drive improved fitness [7,11,12,13,14,15].

It is unknown whether and how stress-induced genome instability in human fungal pathogens is regulated but it is likely that chromatin-mediated epigenetic regulation plays a key regulatory role in this process. Indeed chromosome rearrangements do not occur randomly and are more likely to occur at specific chromosomal locations enriched in DNA repeats [6,13,16,17]. Unequal exchange between homologous repeats can alter the length of the repetitive region and inter- and intra-chromosomal recombination can result in gene conversion, gross chromosomal rearrangements and segmental aneuploidies. In many organisms, assembly of heterochromatin promotes genome stability by repressing inappropriate recombination and transposon activity [18,19].

Despite our detailed knowledge of heterochromatin structure and function in fungal model systems, such as *Saccharomyces cerevisiae* and *Schizosaccharomyces pombe*, we still know little about the structure and function of repeat-associated chromatin in human fungal pathogens. 

In this review, I use our knowledge of heterochromatin structure and function in fungal model systems as a road map to review the role of chromatin in regulating genome plasticity in human fungal pathogens.

## 2. Regulation of Chromatin Structure by Post Translation Modification of Histone Proteins

The eukaryotic genome is packaged into chromatin, which is composed of basic units called nucleosomes. Each nucleosome comprises two molecules each of histone H2A, H2B, H3 and H4, which form a spool to wrap 147 bp of DNA [20].

Histones are globular proteins composed of a histone core, which mediates interactions between different histone proteins, and unstructured amino terminal tails, which extend out from the core nucleosome particle. Histone tails are subject to a wide variety of post-translational modifications, known as histone marks, that decorate distinct genomic regions and reflect different functional states [21]. The enzymes responsible for adding or removing these epigenetic marks are often termed “writers” and “erasers”, respectively (Figure 2). Histone marks affect chromatin structure via two distinct mechanisms: (i) they can directly influence histone–histone and histone–DNA interactions or (ii) they can be bound by “reader” proteins (Figure 2). Readers recruit various components of the nuclear signalling network to chromatin, mediating fundamental nuclear processes such as transcription, DNA replication and recombination [21]. 

Chromatin can exist in two states: euchromatin and heterochromatin. Euchromatin assembles over gene-rich, non-repetitive DNA and it is associated with “active” histone marks permissive to transcription such as histone acetylation and methylation of lysine 4 on histone H3 (H3K4me) [18]. 

Two different types of heterochromatin are assembled into distinct genomic regions. Constitutive heterochromatin (here simply termed heterochromatin) assembles at loci with low gene density and/or enriched in repetitive DNA where it silences the expression of most genes within or near it and promotes genome stability by suppressing unequal recombination events [22]. Facultative heterochromatin is a chromatin state that can reversibly adopt an open or compact conformation and is found at developmentally-regulated genes or transcriptionally-silent genes that can be activated under specific circumstances [21].

In mammalian cells, the most prominent chromatin feature of heterochromatin is global hypoacetylation, which leads to chromatin fibre compaction. In addition, specific “repressive” marks flag heterochromatic regions. A typical mark of heterochromatin is the methylation of histone H3 on lysine 9 (H3K9me) established by the H3K9 methyltransferases Su(var)3-9 [23]. Heterochromatic regions are also flagged by DNA that is methylated on the 5- position of cytosine (5mC) due to the action of establishment and maintenance DNA methyltransferases [24]. Facultative heterochromatin is marked by H3K27 methylation established by the methyltransferase Ezh2 [25]. 

## 3. Fungal Model Systems as a Road Map for Understanding Heterochromatin Structure and its Function in Promoting Genome Stability

The histone modification state associated with euchromatin is largely conserved across fungal organisms. In contrast, the chromatin states associated with heterochromatic regions differ between fungal organisms including the two well-studied model systems *S. cerevisiae* and *S. pombe* (Figure 3).

### 3.1. Sir2-Dependent Heterochromatin in Saccharomyces cerevisiae

*S. cerevisiae* (Saccharomycetaceae family, Saccharomycotina subphylum of Ascomycota) arose from complete duplication of eight ancestral chromosomes followed by massive gene loss (nearly 90% of duplicated genes) and gene specialization (Figure 3) [26,27]. *S. cerevisiae* is a well-established model for species as diverse as humans and pathogenic fungi and it has been used by humans for thousands of years to produce food and drink products that rely on fermentation. However, *S. cerevisiae* is also isolated clinically and it is an emerging opportunistic pathogen [28].

In *S. cerevisiae*, heterochromatin (also known in this organism as “silent chromatin”) lacks most repressive histone marks, which are normally associated with mammalian heterochromatin. For example, the *S. cerevisiae* genome does not encode for Su(var)3-9 or Ezh2 orthologues and therefore its chromatin lacks the H3K9me and H3K27me repressive marks [29]. *S. cerevisiae* has also no endogenous DNA methylation machinery and it is therefore devoid of 5mC DNA methylation [29]. The most prominent feature of *S. cerevisiae* heterochromatin is the lack of active histone marks—most importantly histone H3 lysine 16 acetylation (H3K16Ac; Figure 4).

This hypoacetylated histone state is ensured by the coordinated action of chromatin erasers and readers where deacetylation by the NAD^+^-dependent histone deacetylase Sir2 promotes binding of the readers Sir3 and Sir4, allowing further recruitment of Sir proteins to assemble heterochromatin [30,31].

*S. cerevisiae* heterochromatin might also be flagged for DNA damage as phosphorylation of serine 129 on histone H2A (known as γH2A), a hallmark of DNA double-stranded breaks, is enriched at heterochromatin regions [32,33]. In contrast, in mammalian cells, phosphorylation of H2AX, a modification functionally analogous to the γH2A modification, does not mark heterochromatin [34].

As seen in other organisms, *S. cerevisiae* heterochromatin assembles over genomic locations enriched in repetitive DNA: the silent mating-type (*MAT*) locus, subtelomeric regions and ribosomal DNA tandem arrays [35]. The role of heterochromatin in regulating gene expression and genome stability at these regions has been intensively explored [36,37,38,39,40,41].

#### 3.1.1. Heterochromatin at the *S. cerevisiae* the *MAT* Locus

Heterochromatin is assembled at the *MAT l*ocus where, by controlling gene expression and recombination, it allows faithful mating-type switching, enabling sister cells to mate to produce a diploid. The *MAT* locus directs mating-type switching: a biological switch that allows haploid cells of one mating type to produce haploid cells of the other mating type [42]. The *MAT* locus contains either the a1 gene (*MATa*) or the α1 and α2 genes (*MAT*α). These genes encode transcription factors that determine the expression of cell-type-specific genes necessary for pheromone production and detection. *MAT* shares homology with two other sites located on the same chromosome: *HML*α and *HMR*a. Both loci have complete copies of the mating-type genes, but are kept transcriptionally silent by Sir2-dependent heterochromatin. Accordingly, haploid cells defective in heterochromatin express both sets of mating type genes at once and therefore are mating defective [39,40]. 

In wild-type (WT) cells mating-type switching is triggered by the HO site-specific endonuclease that specifically cleaves the *MAT* locus. This double-strand break is repaired by homologous recombination using the *HML*α and *HMR*a loci as a donor, resulting in a unidirectional gene conversion event where the α or a gene from the *HML/HMR* locus is copied into the *MAT* locus. The recognition sequence for the HO endonuclease is present not only at the *MAT* locus but also at the *HML* and *HMR* loci. However, heterochromatin assembled at *HML* and *HMR* blocks access of the HO endonuclease to these loci, so cutting occurs only at *MAT* [37,43]. Therefore, at the *MAT* locus, heterochromatin promotes site-specific homologous recombination in addition to repressing gene expression at the *HML* and *HMR* loci.

#### 3.1.2. Heterochromatin at the *S. cerevisiae* Subtelomeres

Subtelomeres are gene-poor and repeat-rich regions found adjacent to telomeric repeats at both ends of each linear chromosome [44]. In *S. cerevisiae*, subtelomeric regions contain several copies of X and Y’ non-coding DNA elements and are enriched for fast-evolving multi-gene families [44,45]. Subtelomeric gene families are often involved in niche adaptation and specialised metabolic functions that reflects the lifestyle of *S. cerevisiae* [44,46].

Subtelomeric repeats are assembled into Sir2-dependent heterochromatin. The role of subtelomeric heterochromatin in *S. cerevisiae* remains a mystery as its role in transcriptional repression and inhibition of genetic recombination seem to be minor. Indeed, even though subtelomeric heterochromatin can repress gene expression of marker genes integrated in proximity of certain X elements, only 6% of endogenous subtelomeric genes are silenced in a Sir-dependent manner [36,38]. In addition, *S. cerevisiae* subtelomeric heterochromatin does not seem to play a major role in repression of mitotic recombination. Instead, genome stability at subtelomeres is promoted by the Ku70-Ku80 heterodimer, a protein complex involved in non-homologous end-joining and telomere end maintenance [29].

#### 3.1.3. Heterochromatin at the *S. cerevisiae* rDNA Locus

The ribosomal DNA (rDNA) locus contains genes encoding for ribosomal RNAs (rRNAs) consisting of a tandem array of a ~9.1 kb unit repeated 100 to 200 times on chromosome XII. Each unit contains the 35S rRNA gene that is transcribed by RNA polymerase I and the 5S rRNA gene that is transcribed by RNA polymerase III. The 35S and 5S genes are separated by two non-transcribed spacers (NTS), also known as intergenic spacers (IGS), that contain three conserved elements: (i) an origin of replication (rARS), (ii) a replication fork blocking site (RFB) whose function is to avoid collision between replication and transcription machineries and (iii) a non-coding RNA promoter (E-pro) whose activity can be silenced by Sir2 [41,47,48]. The activity of these conserved elements and their chromatin state is critical to maintain an optimal number of rDNA repeats that, due to their repetitive nature, are highly unstable genome sites. Indeed, the length of the rDNA locus is always changing in a cyclic manner, as events that cause rDNA amplification alternate with those that lead to rDNA contraction. rDNA amplification is triggered by DNA replication arrest at the RFB site. An arrested DNA replication fork induces a double strand break and DNA repair by unequal sister-chromatid recombination will result in an increase in the number of rDNA repeats [41]. 

rDNA contraction is promoted by Sir2-dependent heterochromatin via repression of transcription from the E-pro promoter [41]. In the absence of Sir2 high levels of transcription from E-pro enhance rDNA recombination. Therefore, Sir2-dependent heterochromatin plays critical roles in repressing excessive rDNA amplification thereby promoting genome stability. 

### 3.2. Heterochromatin in Schizosaccharomyces pombe 

The fission yeast *Schizosaccharomyces pombe* is a representative of the Taphrinomycotina subphylum of Ascomycota (Figure 3). Phylogenetic analyses suggest that Taphrinomycotina and the Saccharomycotina subphyla diverged around 540 million years (Myr) ago [49]. *S. pombe* is an important fungal model system in which to dissect heterochromatin assembly and function because its architecture closely resembles that of Metazoa. Like *S. cerevisiae*, *S. pombe* lacks DNA methylation and the H3K27me system [50]. However, similar to mammalian cells, *S. pombe* heterochromatin is typified by hypoacetylated nucleosomes that are methylated on H3K9. The coordinate action of the RNA interference (RNAi) machinery, chromatin writers and readers establishes and maintains this chromatin state (Figure 5). Repetitive DNA elements are transcribed into non-coding double-stranded RNAs (dsRNA) and processed into small interfering RNAs (siRNAs) by the RNAi machinery [51]. siRNAs seed heterochromatin assembly by recruiting the protein complex containing the writer histone-H3K9 methyltransferase Clr4/Su(var)3-9. H3K9me is a binding site for chromodomain reader proteins such as Swi6/HP1, allowing heterochromatin assembly and spreading [23,52,53]. The writer Clr4/Su(var)3-9 also contains a chromodomain and therefore it also reads H3K9me, facilitating heterochromatin maintenance through cell division [54]. The hypoacetylated state of fission yeast heterochromatin is maintained by the action of histone deacetylases: Sir2, Clr3 and Clr6 (Figure 5) [55,56,57].

*S. pombe* heterochromatin is found mainly at repeat-rich genomic locations: pericentromeres, the *MAT* locus, subtelomeres and the rDNA locus where it regulates genome stability via different mechanisms. In addition, islands of facultative heterochromatin are detected across the genome.

#### 3.2.1. Heterochromatin at *S. pombe* Pericentromeres 

Whereas *S. cerevisiae* contains small (125 bp) point centromeres, *S. pombe* contains large (35–110 kb) regional centromeres [58]. Fission yeast centromeres are composed of a central non-repetitive domain upon which the kinetochore assembles, flanked by outer-repeat sequences (*otr*, consisting of *dg* and *dh* repeats) coated in heterochromatin [53,59,60]. Pericentromeric heterochromatin is important for faithful chromosome segregation as it ensures high levels of cohesion to hold sister chromatids together when confronted with tension from the spindle microtubules [61]. As a consequence heterochromatin-null mutant strains display defective chromosome segregation, an elevated number of lagging chromosomes and sensitivity to the microtubule destabilizing drug thiabendazole (TBZ) [62]. Pericentromeric heterochromatin also inhibits deleterious homologous recombination events that could arise from collisions between the transcriptional and replication machinery [63], and represses centromeric meiotic recombination [64].

#### 3.2.2. *S. pombe* Heterochromatin at Other Genomic Locations

In addition to pericentromeric regions, the *S. pombe MAT* with its two silent cassettes, subtelomeric regions and the *rDNA* locus are assembled into transcriptionally silent heterochromatin that promotes genome stability. At the *MAT* locus, heterochromatin promotes mating-type switching by using a mechanism similar to that operating in *S. cerevisiae* [53], subtelomeric heterochromatin plays a role in pairing of homologous chromosomes in meiotic prophase and hence chromosome segregation [65] while rDNA heterochromatin is instrumental in cell survival following telomere loss [66].

In addition to the large block of heterochromatin found at centromeres, the mating-type region, rDNA, and telomeres, smaller “heterochromatic islands” can be found across the *S. pombe* genome. These heterochromatin islands reversibly regulate gene expression in response to environmental changes. For example, during vegetative growth heterochromatic islands assemble at meiotic genes and are important for their transcriptional silencing [67], and other heterochromatin islands regulate expression of environmentally controlled transcripts [68].

## 4. Heterochromatin Structure and Function in Human Fungal Pathogens

### 4.1. Distinct Chromatin States Mark Different Repetitive Elements in Candida albicans

The yeast *C. albicans* belongs to the “CTG-clade” (Debaryomycetaceae family, Saccharomycotina subphylum of Ascomycota; Figure 3). Members of the “CTG-clade” diverged from the Saccharomycetaceae more than 230 Myr and they are characterized by an alternative genetic code in which the CUG codon encodes the amino acid serine instead of leucine [69,70]. *C. albicans* is an opportunistic fungal pathogen that is part of the normal microflora of most healthy individuals. However, in immunocompromised patients, this benign commensal organism becomes a life-threatening pathogen, causing systemic infections that are fatal in ~50% of cases [71]. Such infections are the fourth most common infection in hospitals and are treated with anti-fungal drugs, often leading to drug resistance [2]. *C. albicans* is such a successful pathogen because it can colonise almost every organ in the human body and it can efficiently adapt to different hostile host-niche environments.

In the last decade, it became clear that genome plasticity is central to *C. albicans* adaptation. *C. albicans* is a diploid organism with a genome organised into 2 × 8 chromosomes (2n = 16). However, clinical isolates exhibit karyotypic diversity, including aneuploidy and gross chromosomal rearrangements that can confer anti-fungal drug resistance due to altered copies of specific genes [11,12,13]. Experimental evolution has demonstrated that *C. albicans* genome instability is induced by environmental conditions: the genome is relatively stable under optimal laboratory growth conditions but becomes more unstable under stress conditions. For example, chromosome rearrangements and/or aneuploidy dramatically increased in the presence of the anti-fungal drug fluconazole (100 fold) or at high temperature (10 fold) [9]. Genome instability also increases during infections of *in vivo* mouse models [16,72,73]. Given that *C. albicans* is mainly an asexual organism lacking meiosis, genomic diversity is produced by mitotic genome instability [74].

Different types of DNA repeats are the major source of genome instability. These DNA elements include: (i) long, complex repetitive elements [75], (ii) long terminal repeats (LTR) and Zorro non-LTR retrotransposons [76], (iii) short repeat sequences such as short tandem repeats and trinucleotide repeats and (iv) long (65–6499 bp) inverted repeats [17]. 

Among the different types of repetitive DNA elements, long, complex repetitive elements are potentially perfect substrates for heterochromatin assembly. Long, complex repetitive elements are found at telomeric regions, the rDNA locus, and the major repeat sequence (MRS) [75]. Telomeres are composed of tandemly repeating 23-bp units, while subtelomeres are enriched in long terminal repeats (LTR), retrotransposons, and gene families, such as the *TLO* genes [75,77]. The rDNA locus, located on chromosome R, has an overall architecture similar to the *S. cerevisiae* rDNA locus and consists of a tandem array of a ~12 kb unit repeated 50 to 200 times [75]. MRS loci are a class of *Candida*-specific DNA repeats composed of long tracts (10–100 kb) of nested DNA repeats that are found on 7 of the 8 *C. albicans* chromosomes [78]. The *C. albicans MAT* locus does not contain the *S. cerevisiae*-like *HML* and *HMR* cassettes for mating-type switching. In contrast *C. albicans*, possesses a single mating-type (MTL) locus on chromosome 5, which is normally heterozygous (a/α) in this diploid organism [79]. 

Similarly to *S. cerevisiae*, the *C. albicans* genome encodes for the histone deacetylase Sir2 but it lacks the H3K9 and H3K27 methylation machinery [50]. 5mC DNA methylation has been detected in *C. albicans,* although DNA methyltransferase orthologues cannot be identified by bioinformatic analyses [50]. Finally, the *C. albicans* genome encodes for components of the RNAi machinery (Figure 6) [80]. The role of *C. albicans* RNAi is controversial: while siRNA sequencing supports the presence of an active RNAi pathway, RNAi in *C. albicans* is not triggered by exogenously introduced dsRNA, a common technique used to study RNAi [80,81]. It is unknown whether *C. albicans* RNAi contributes to chromatin structure.

Analysis of the chromatin state associated with the different types of complex repetitive elements has demonstrated that each repetitive element is associated with a distinct chromatin state.

Sir2-dependent heterochromatin assembles at subtelomeric regions and the rDNA locus where it dynamically represses expression of subtelomeric genes, leading to transcriptional noise and repression of non-coding RNAs originating for the non-trascribed spacer (NTS) regions of the rDNA locus [82,83,84]. While *C. albicans* Sir2 is largely dispensable for repressing recombination at the rDNA locus, Sir2 represses subtelomeric stability by acting on the subtelomeric recombination hotspot TLO recombination element (TRE) [10]. The DNA damage hallmark γH2A is also enriched across subtelomeric regions and at the rDNA locus, suggesting that these regions undergo frequent DNA damage [83]. It is unknown whether and how these γH2A-enriched regions are continuously repaired and whether DNA damage repair at these sites contributes to *C. albicans* genome instability. DNA that is methylated on cytosine (5mC) has been detected at the rDNA locus and subtelomeric regions, suggesting that DNA methylation is an important contributor to *C. albicans* heterochromatin structure [85]. It will be important to identify the DNA methyltransferase(s) that establish and maintain this modification as this will allow dissection of the role of DNA methylation in the regulation of gene expression and genome stability in *C. albicans*.

Given their highly repetitive nature, MRS repeats are expected to be ideal substrates for heterochromatin assembly. Surprisingly, Sir2-dependent hypoacetylated heterochromatin does not mark the MRS repeats. MRSs are assembled into highly acetylated chromatin that is permissive to transcription [82,83]. However, MRS repeats are associated with nucleosomes that lack the active histone mark H3K4 methylation [82,83]. It is still unknown whether the hypomethylation on H3K4 contributes to MRS stability. Lack of heterochromatin at MRSs could also explain the instability of these repetitive elements. Indeed, analyses of clinical isolates and experimental evolution with an *in vivo* mouse model system has demonstrated that in the host MRS are unstable genomic loci as they can expand and contract and are known sites of translocations [16,86].

It will be important to determine whether and how the chromatin state associated with *C. albicans* repetitive elements is remodelled upon entry into hostile environments such as those encountered during colonisation and infection of the host.

### 4.2. Heterochromatin as a Regulator of Secondary Metabolite Gene Clusters in Aspergillus fumigatus 

The *Aspergillus* genus belongs to the Pezizomycotina subphylum (Figure 3). Pezizomycotina and Saccharomycotina diverged ~530 Myr [87]. The *Aspergillus* genus is extremely diverse, consisting of almost 200 species of ubiquitous, saprophytic fungi, including species useful in industrial processes, genetic model systems as well as opportunistic fungal pathogens. Among the human pathogenic species of *Aspergillus*, *A. fumigatus* is responsible for 90% of human infections, followed by *A. flavus*, *A. terreus*, *A. niger* and the model organism *A. nidulans* [88]. Human Aspergillosis can range from allergic syndromes to invasive aspergillosis, a disease that has a mortality rate of up to 90%. The increasing incidence of anti-fungal resistance is a growing concern [89].

*A. fumigatus* is an opportunistic filamentous fungus whose natural ecological niche is the soil, food and decaying vegetation where it grows on organic debris. In the soil, it sporulates, forming thousands of airborne spores, known as conidia, able to survive in a wide range of environments. The conidia are small enough (2 to 3 μm) to reach the lung alveoli. Inhalation of infectious conidia by immunocompromised individuals may lead to germination and growth of the fungus in the lung, followed by progression to serious diseases such as invasive pulmonary aspergillosis. Following inhalation, *A. fumigatus* adheres and penetrates the human respiratory epithelia and overcomes the response of surrounding cells particularly phagocytic cells, and therefore it must survive in a variety of host-niches environments.

*A. fumigatus* has a haploid genome and possesses a fully functional sexual reproductive cycle [90]. Mating is controlled by the *MAT* locus that, similarly to *C. albicans* and *C. neoformans*, does not possess a *S. cerevisiae*-like silent-cassette system.

Genetic diversity associated with subtelomeric regions might be important for driving adaptation to different host niche environments. Subtelomeric regions are the most diverse genomic regions across different *A. fumigatus* isolates and harbour many gene clusters required for synthesis of secondary metabolites (SM): small molecules that are not essential for life but are important virulence factors [91,92,93]. Indeed, sequencing of 66 *A. fumigatus* strains revealed an incredible intra-species genome variation of SM gene clusters with SNPs, gene deletions, gene amplifications and reciprocal translocations [94]. 

Although heterochromatin structure and function in *A. fumigatus* is still largely unexplored, studies performed in *A. nidulans* and other *Aspergillus* species strongly suggest that heterochromatin is a critical regulator of subtelomeric SM gene clusters.

Heterochromatin in *Aspergillus* is marked by hypoacetylated nucleosomes that are methylated on H3K9, while no H3K27 methylation system has been identified [50,95]. It is unlikely that DNA methylation is a major contributor to heterochromatin structure. Indeed, although earlier studies reported the presence of DNA methylation in *A. flavus* [96], subsequent studies failed to detect any DNA methylation in this organism [97,98]. *Aspergillus* species contain an active RNAi machinery that has an anti-viral defence role but it is unknown whether RNAi contributes to heterochromatin formation (Figure 7) [99,100].

In *A. nidulans*, transcription of subtelomeric SM gene clusters is repressed during the active growth phase and activated upon growth arrest. The transcriptionally repressed state during the active growth phase is mediated by heterochromatin that is hypoacetylated and methylated on H3K9. Accordingly, deletion of the H3K9 methyltransferase ClrD/Su(var)3-9 and of the histone deacetylase Hda1 leads to increased levels of SM in cycling cells [101,102]. *A. fumigatus* contains an active H3K9 methyltransferase ClrD/Su(var)3-9 and an orthologue of Swi6/HP1 (*Afu6g07550*). Deletion of the *A. fumigatus CLRD* gene results in developmental defects but its contribution to subtelomeric secondary metabolite gene clusters expression has not yet been investigated [103]. It will be of particular interest to investigate whether and how *A. fumigatus* heterochromatin structure contributes to SM gene expression regulation and the genome plasticity of SM gene clusters. 

### 4.3. Distinct Chromatin States Associated With Cryptococcus Neoformans Repetitive Elements

*Cryptococcus neoformans* is a pathogenic basidiomycetous yeast responsible for more than 600,000 deaths annually worldwide (Figure 3) [104]. Basidiomycota and Ascomycota diverged ~700 Mya [105]. The burden of cryptococcal diseases is remarkably high in immunocompromised patients, especially in human immunodeficiency virus (HIV) HIV-infected individuals [104]. The increasing resistance to the few available anti-fungal drugs is a growing public health threat that should be tackled with a matter of urgency [2].

*C. neoformans* is mainly found in the environment and it penetrates the pulmonary alveoli following inhalation of spores or desiccated yeast cells. This allows its dissemination through the bloodstream causing life-threating infections, most often meningoencephalitis [106]. 

Genome instability is likely to be a key driver of *C. neoformans* pathogenicity and of its ability to survive in different hostile host niches. Indeed, upon exposure to the host lung, *C. neoformans* produces large polyploid titan cells. While typical cryptococcal cells are haploid and have a diameter of 5 to 7 µm; titan cells are predominantly tetraploid or octoploid and can be 5 to 10 times larger than normal cells [107]. *C. neoformans* has a sexual cycle that allows mating of two haploid cells of different mating type. However, titan cell formation does not depend on mating but it is induced by specific environmental growth conditions [108,109,110]. Titan cells are required for survival within the mouse host and for causing disease [107]. This is probably due to a combination of factors. First of all, the large size and the cell wall composition of titan cells leads to reduced phagocytosis by the cells of the host immune system [111]. In addition, the polyploid state of Titan cells enhances the ability of *C. neoformans* to adapt to stress conditions and therefore it might enable survival within different hostile host niches [15]. This is because polyploid genomes are highly unstable and the offspring of polyploid cells frequently carry diverse genomic alterations, allowing selection of genotypes better suited to grow in stress environments. Indeed, genome instability (especially aneuploidy) is instrumental for the development of antifungal drug resistance in *C. neoformans* [5,112,113].

It is still unknown whether the chromatin state associated with repetitive elements of *C. neoformans* regulates genome stability in typical *C. neoformans* or Titan cells.

The *C. neoformans* genome encodes many heterochromatin components that are associated with mammalian heterochromatin (Figure 8). Indeed, *C. neoformans* encodes orthologues of the H3K9 methyltransferase Clr4/Su(var)3-9 and of the H3K27 methyltransferase Ezh2. In addition, *C. neoformans* possesses 5mC DNA methylation that is dependent on the sole cytosine DNA methyltransferase Dnmt5 [114,115] and it has an active RNAi machinery [116]. Combinations of these heterochromatic marks are found at centromeres and subtelomeres. Heterochromatin does not decorate the mating-type locus because *C. neoformans* lacks silent mating-type cassettes and does not switch mating type [117].

*C. neoformans* contains large regional centromeres (20–40 kb) that are gene-poor and enriched in retrotransposons [118,119,120]. Centromeres are assembled into a chromatin-state containing all the markers of constitutive heterochromatin where histones are hypoacetylated and methylated on H3K9. The chromodomain proteins Swi6/HP1 and Dnmt5 are the H3K9me chromatin readers that target 5mC DNA methylation at centromeres [115]. The role of centromeric heterochromatin in the regulation of faithful chromosome segregation has not yet been explored. Centromeric retrotransposons are targeted by the RNAi machinery for siRNA production and, in the absence of RNAi, centromeres are prone to structural alterations [119]. This suggests that RNAi ensures genome stability at centromeres by suppressing mitotic recombination. However, RNAi probably does not contribute to centromere function in *C. neoformans* as, unlike what is observed in *S. pombe*, RNAi null mutant strains are not sensitive to the microtubule destabilising drug TBZ, indicating that faithful chromosome segregation does not require the RNAi machinery [116]. Finally, it is not yet established whether RNAi is required for centromeric heterochromatin establishment and maintenance, and it seems unlikely given that *C. neoformans* RNAi-mediated gene silencing, such as sex-induced silencing, is post-transcriptional and is not associated with H3K9 methylation [121].

A different combination of heterochromatic marks is detected at subtelomeric regions. Low levels of H3K9me can be detected at subtelomeres and H3K9me is probably responsible for targeting 5mC DNA methylation to these regions. In contrast, high levels of H3K27 methylation, a chromatin mark that is in general associated with facultative heterochromatin, is detected at subtelomeres [114,115,122]. The protein complex PRC2 is responsible for depositing the H3K27 methylation mark as it contains the H3K27me chromatin writer Ezh2 and the H3K27me chromatin reader Ccc1 [122]. Histone hypoacetylation cooperates with H3K27 methylation to maintain transcriptionally repressive subtelomeric chromatin structure as the orthologue of the histone deacetylase Clr3 cooperates with the H3K27 methylation machinery to silence gene expression of embedded genes [122]. It is unknown whether subtelomeric heterochromatin controls genome stability of subtelomeric regions. 

Thus, different types of heterochromatin assemble at *C. neoformans* centromeric and telomeric regions and it will be important to investigate whether and how these chromatin states contribute to genome instability in canonical and titan cells.

## 5. Advancing Our Understanding of Heterochromatin Structure and Function in Human Fungal Pathogens

Data from many different laboratories have highlighted how heterochromatin structure is different in different fungal organisms. Given that heterochromatin is likely to play key regulatory roles in stress adaptation, there is a real need to investigate the chromatin state associated with DNA repeats in human fungal pathogens. Investigation of heterochromatin structure and function in this group of medically relevant microorganisms is facilitated by the robust methods developed in the fungal model systems *S. cerevisiae* and *S. pombe,* which can be easily adapted for human fungal pathogens.

Genome-wide maps of active and repressive histone marks have been instrumental in identifying heterochromatin regions across eukaryotic genomes [36,123]. Such epigenomic maps can be obtained by performing chromatin immunoprecipitation followed by high-throughput sequencing (ChIP-seq) using antibodies that specifically recognize the histone marks. Given that histone proteins are largely conserved across eukaryotes, these experiments can be performed using histone antibodies whose specificity has been tested in *S. cerevisiae* and *S. pombe*. To distinguish between nucleosome occupancy and depletion/enrichment of specific histone modifications, ChIP-seq should also be conducted with antibodies that recognize unmodified histones. Similarly, 5mC DNA methylation can be detected at single base-pair resolution by bisulfite genomic sequencing [124].

ChIP-seq has one major drawback: it requires a priori knowledge of the type and position of the histone modification of interest. It is possible that repeat-rich genomic regions of human fungal pathogens are assembled into a chromatin state marked by novel histone modification(s). Novel histone modifications could be identified by performing mass spectrometry (MS)-based proteomics [125]. ChIP-seq and MS are complementary strategies as MS analysis of bulk chromatin preparation limits the identification of histone modification to a global view and does not provide information about their patterning in distinct genomic locations. Recently, methodologies coupling ChIP with MS have been developed to study the composition of chromatin-associated complexes and further development of these methodologies could be instrumental to advance our understanding of heterochromatin composition in human fungal pathogens [125].

Heterochromatin function can also be investigated by adapting simple and affordable genetic systems to human fungal pathogens. The transcriptionally repressive state of heterochromatin regions can be assessed by developing reporter gene silencing assays where a reporter gene (such as *URA3^+^*) is integrated at heterochromatic regions [126] Transcriptional silencing of these reporter genes can be assessed by plating serial dilution of cells to selective media. For example, transcriptional repression of the *URA3^+^* marker gene renders cells resistant to 5-fluoroorotic acid (5-FOA). Similarly, genomic instability at repeat-rich regions can be assessed genetically by for example measuring loss of heterozygosity (LOH) rates of a heterozygous marker gene inserted at repeat-rich regions or by measuring gene conversion between heteroalleles of an easily scored nutritional marker [9,127]. These genetic approaches can be easily developed in human fungal pathogens as, in the past decade, several crucial tools have greatly enhanced our ability to manipulate human fungal pathogens genetically.

## 6. Concluding Remarks

Chromatin structure and function in human fungal pathogens is still a poorly investigated field of research. However, the limited analyses that have been performed have highlighted how heterochromatin is often atypical in human fungal pathogens as the basic building blocks of heterochromatin are rearranged in this diverse group of medically important microorganisms. Given that similar structure often equates to different functions, it is dangerous to directly extrapolate findings from yeast model systems to human fungal pathogens. Heterochromatin and its ability to rapidly and reversibly regulate gene expression and genome instability could be a key regulatory mechanism underlying virulence and pathogenicity of human fungal pathogens. It is therefore surprising that we still know very little about chromatin structure and function in this medically important microorganisms and future research will be instrumental in unveiling its role in virulence and pathogenesis.

## Figures and Tables

**Figure 1 genes-10-00855-f001:**
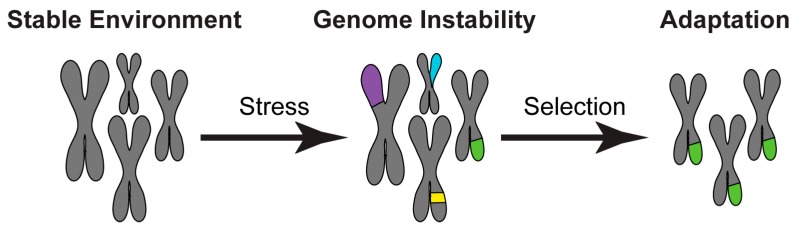
Stress-induced genome instability as a key adaptive mechanism. In a stable environment, the genomic organisation (grey) of microbial organisms allows optimal growth. Environmental changes are sensed as a stress and induce genome instability. The increased genetic diversity allows selection of genotype(s) (grey and green) that are better adapted in the new environment.

**Figure 2 genes-10-00855-f002:**
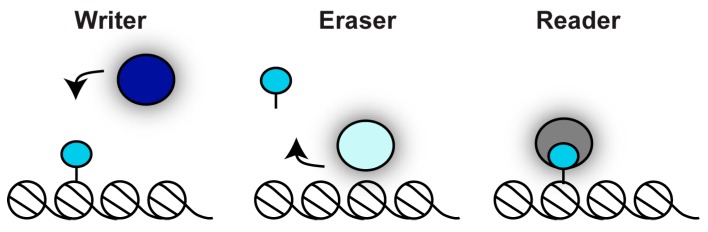
Chromatin writer–eraser–reader model. A chromatin mark (blue circle) is established by a writer enzyme and erased by an eraser enzyme. The chromatin mark is recognised by a specific binding protein: the reader.

**Figure 3 genes-10-00855-f003:**
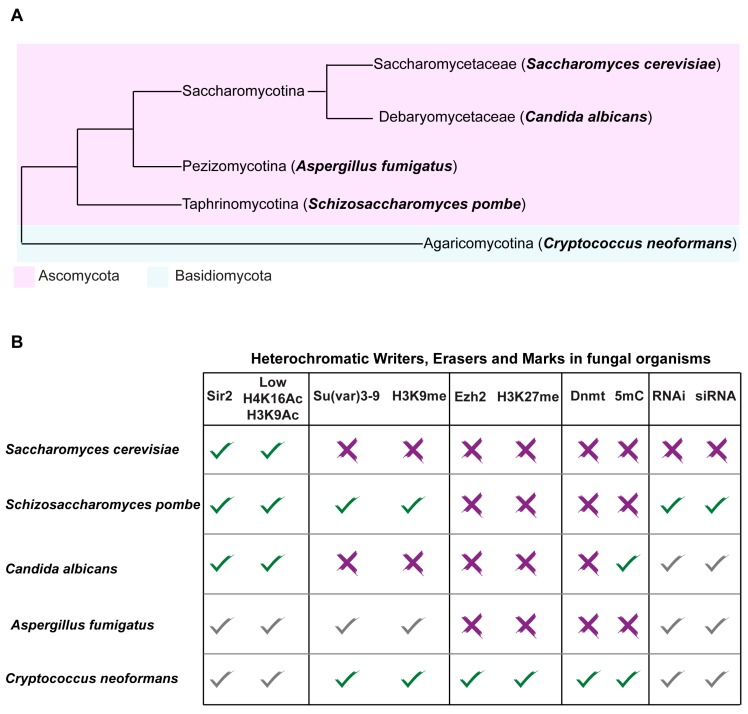
Heterochromatin structure differs between fungal organisms. (**A**) Phylogenetic tree of the fungal organisms discussed in this review; (**B**) presence and absence of selected writers, erasers and marks associated with heterochromatic regions in the fungal model systems *Saccharomyces cerevisiae* and *Schizosaccharomyces pombe* and in the human fungal pathogens *Candida albicans*, *Aspergillus fumigatus* and *Cryptococcus neoformans*. Green check: experimentally validated action of writers, erasers and marks at heterochromatic regions; Red cross: lack of orthologues/absence of marks at heterochromatic regions; and grey check: presence of orthologous genes but lack of experimental data demonstrating action of writers, erasers and histone marks at heterochromatic regions.

**Figure 4 genes-10-00855-f004:**
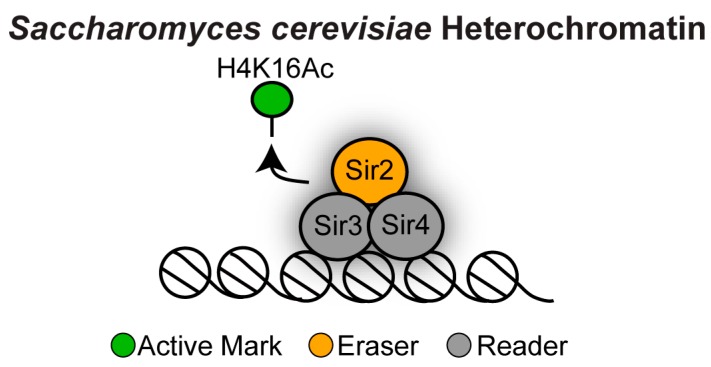
Heterochromatin structure in *Saccharomyces cerevisiae*. The NAD^+^-dependent histone deacetylase Sir2 erases H4K16 acetylation (H4K16Ac) promoting binding of the readers Sir3 and Sir4.

**Figure 5 genes-10-00855-f005:**
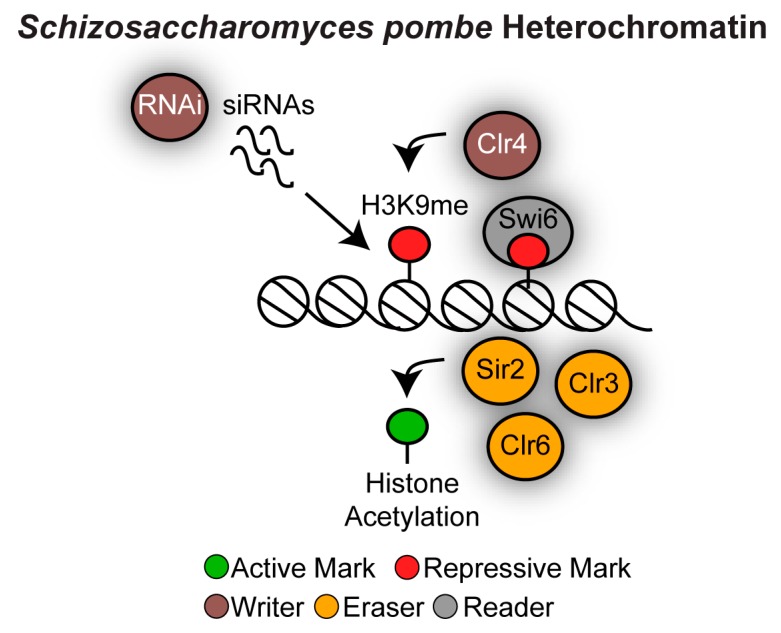
Heterochromatin structure in *Schizosaccharomyces pombe*. The RNAi machinery seeds heterochromatin assembly by recruiting the protein complex containing the writer histone-H3K9 methyltransferase Clr4. H3K9me is a binding site for chromodomain reader proteins such as Swi6. The histone deacetylases Sir2, Clr3 and Clr6 maintain the hypoacetylated histone state.

**Figure 6 genes-10-00855-f006:**
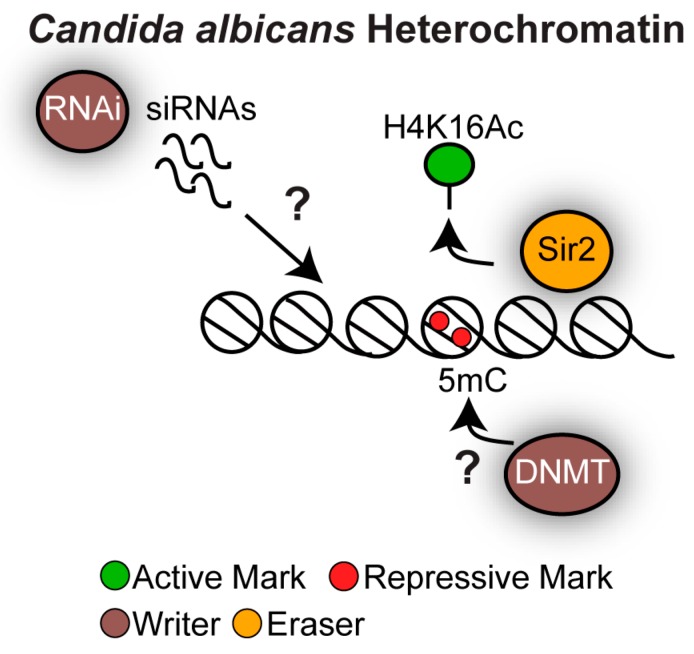
Heterochromatin structure in *Candida albicans*. The NAD^+^- dependent histone deacetylase Sir2 maintains the hypoacetylated histone states. Repetitive elements are decorated by 5mC DNA methylation by an unknown DNA methyltransferase. It is unknown whether the RNAi machinery contributes to heterochromatin formation.

**Figure 7 genes-10-00855-f007:**
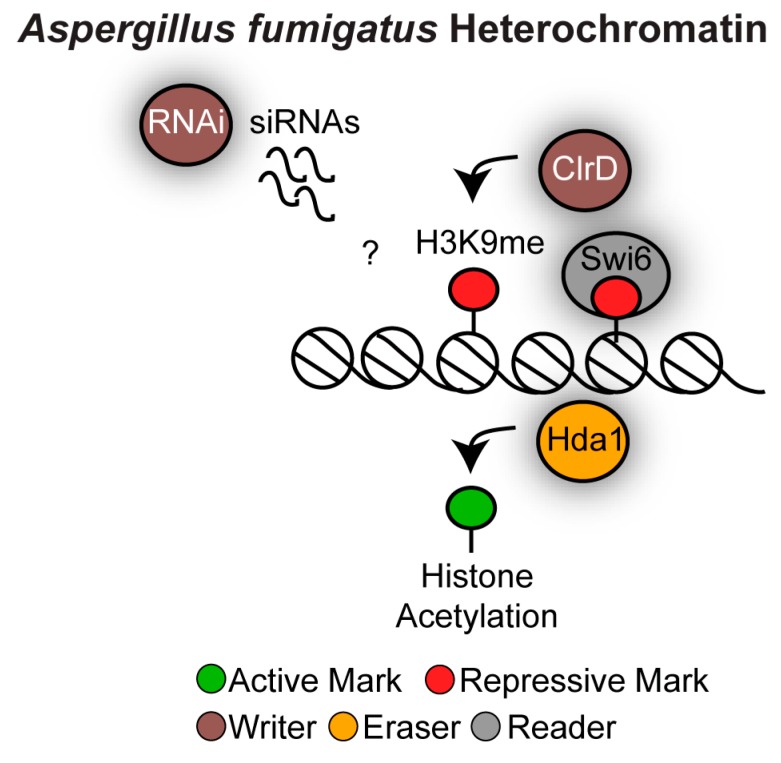
Heterochromatin structure in *Aspergillus fumigatus*. The methyltransferase Clr4. methylates H3K9 creating a binding site for the chromodomain reader proteins Swi6. The histone deacetylase Hda1 maintains the hypoacetylated histone state. It is unknown whether the RNAi machinery contributes to heterochromatin assembly.

**Figure 8 genes-10-00855-f008:**
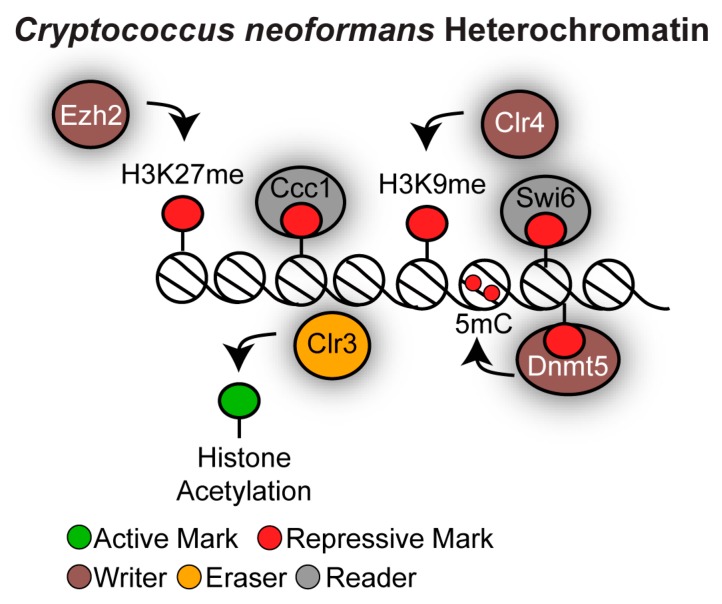
Heterochromatin structure in *Cryptococcus neoformans.* The methyltransferases Clr4 and Ezh3 methylate H3K9 and H3K27 respectively. 5mC DNA methylation that is dependent on the sole cytosine DNA methyltransferase Dnmt5. Swi6 and Dnm5 specifically recognise the H3K9me mark while Ccc1 binds the H3K27me mark. Histone hypoacetylation is ensured by the action of the histone deacetylase Clr3. Centromeres and subtelomeric regions are decorated by distinct histone modifications.

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
