# Peer review of "Chromatin-Mediated Regulation of Genome Plasticity in Human Fungal Pathogens"

_genes, 2019, doi:10.3390/genes10110855_

Round 1

Reviewer 1 Report

Title: Chromatin-mediated regulation of genome plasticity in human fungal pathogens

The author set out to compare and contrast the epigenetic marks and the key players in the regulation of heterochromatin across different human fungal pathogens. My comments below are intended to improve the clarity and organization of the manuscript.

TITLE:

Since the author set out to review the knowledgebase about the heterochromatin-based genome plasticity of human fungal pathogens, the article is appropriately titled.

ABSTRACT:

The abstract succinctly describes the need for and the focus of the review on the heterochromatin structure and function in the most common human fungal pathogens.

MAIN TEXT:

I found the review to be well-written since it clearly presents the readers with the necessary background regarding the regulation of chromatin structure by key histone modification proteins, the existing literature and knowledge detailing how regulation in chromatin structure promotes genome stability in model fungal systems, and how moderate genome instability could facilitate fungal pathogens to rapidly adapt to their new host environment. The author also strikes a good balance between what is known and what needs to be investigated regarding this process in human fungal pathogens. The author has concisely cited relevant references throughout the review except on ‘Line 134’ regarding studies that have explored the gene expression changes and genome stability driven by heterochromatin changes. The review would benefit from including a table listing out the key players of chromatin modifications and their presence/absence in different fungal genomes and their functional differences across the fungal genomes. Lastly, while the author does highlight the aspects of chromatin regulation that are understudied in human fungal pathogens, there is little discussion on what experimental setups, methods, and analyses could be utilized in the future to get insight into some of the missing knowledge about human fungal pathogens. The discussion could be expanded to include the author’s thoughts on ways to improve the knowledgebase.

FIGURES and TABLES:

Figures: Figures 3 – 7 would benefit from using a key differentiating between ‘writers’, ‘erasers’, and ‘readers’. Different colors/shapes could be used for specific enzymes/molecules similar to Figure 2. ‘Line 203’ incorrectly references ‘Fig 3’ instead of ‘Fig 4’.

Tables: As mentioned above, the review would benefit from a summary table that enlists the different key players (e.g. Sir2), their presence / absence in each fungal genome, and their biological function pertaining to chromatin regulation.

Reviewer 2 Report

The manuscript by Alessia Buscaino is an interesting review on a little revised field. I consider that if offers a good reference for researchers that wish to start working with any of the three main human fungal pathogens.

In any case, I think that the review can be improved if a deeper evolutionary comparison of the three pathogens and the two yeasts used as models (S. cerevisiae & S. pombe). I suggest to look deeper into the differences in heterochromatin described for the two model yeasts and the three fungal pathogens. The heterochromatin of  C. albicans, a close relative of S. cerevisiae, is very proximal to that of budding yeast whereas A. fumigatus, another Ascomycota could be, perhaps, closer to S. pombe. Finally, C. neoformans, a Basidiomycota, is a totally different clade. I suggest to add a phylogenetic tree Figure showing the relative positions of these five fungus and human beings to comment then the respective heterochromatin features of each one in relation to it.

Minor points:

It would be useful to comment that there are several known S. cerevisiae pathogenic strains that can be models for C. albicans pathogenicity. I thin the publication recently appeared on centromeric chromatin of fungal pathogens: PLoS Pathog. 2018 Aug 23;14(8):e1007150. doi: 10.1371/journal.ppat.1007150, should be included in the text and references.
